# The Effect of Electrode Topography on the Magnetic Properties and MRI Application of Electrochemically-Deposited, Synthesized, Cobalt-Substituted Hydroxyapatite

**DOI:** 10.3390/nano9020200

**Published:** 2019-02-03

**Authors:** Wei-Chun Lin, Chun-Chao Chuang, Chen-Jung Chang, Ya-Hsu Chiu, Min Yan, Cheng-Ming Tang

**Affiliations:** 1Graduate Institute of Oral Science, Chung Shan Medical University, Taichung 40201, Taiwan; tukust94114wenny@gmail.com (W.-C.L.); a65342976534297@gmail.com (Y.-H.C.); 2Department of Medical Imaging and Radiological Sciences, Chung Shan Medical University, Taichung 40201, Taiwan; jimchao@csmu.edu.tw; 3Chung Shan Medical University Hospital, Taichung 40201, Taiwan; 4Animal Radiation Therapy Research Center, Central Taiwan University of Science and Technology, Taichung 40601, Taiwan; jrchang@ctust.edu.tw; 5Department of Radiological Technology, Central Taiwan University of Science and Technology, Taichung 40601, Taiwan

**Keywords:** Titanium dioxide nanotubes, Cobalt-substituted hydroxyapatite, Magnetic resonance imaging, T_2_-contrast agent, Relaxivity coefficient

## Abstract

Magnetic nanoparticles are used to enhance the image contrast of magnetic resonance imaging (MRI). However, the development of magnetic nanoparticles with a low dose/high image contrast and non-toxicity is currently a major challenge. In this study, cobalt-substituted hydroxyapatite nanoparticles deposited on titanium (Ti-CoHA) and cobalt-substituted hydroxyapatite nanoparticles deposited on titanium dioxide nanotubes (TNT-CoHA) were synthesized by the electrochemical deposition method. The particle sizes of Ti-CoHA and TNT-CoHA were 418.6 nm and 127.5 nm, respectively, as observed using FE-SEM. It was shown that CoHA can be obtained with a smaller particle size using a titanium dioxide nanotube (TNT) electrode plate. However, the particle size of TNT-CoHA is smaller than that of Ti-CoHA. The crystal size of the internal cobalt oxide of CoHA was calculated by using an XRD pattern. The results indicate that the crystal size of cobalt oxide in TNT-CoHA is larger than that of the cobalt oxide in Ti-CoHA. The larger crystal size of the cobalt oxide in TNT-CoHA makes the saturation magnetization (*Ms*) of TNT-CoHA 12.6 times higher than that of Ti-CoHA. The contrast in MRIs is related to the magnetic properties of the particles. Therefore, TNT-CoHA has good image contrast at low concentrations in T_2_ images. The relaxivity coefficient of the CoHA was higher for TNT-CoHA (340.3 mM^−1^s^−1^) than Ti-CoHA (211.7 mM^−1^s^−1^), and both were higher than the commercial iron nanoparticles (103.0 mM^−1^s^−1^). We showed that the TNT substrate caused an increase in the size of the cobalt oxide crystal of TNT-CoHA, thus effectively improving the magnetic field strength and MRI image recognition. It was also shown that the relaxivity coefficient rose with the *Ms*. Evaluation of biocompatibility of CoHA using human osteosarcoma cells (MG63) indicated no toxic effects. On the other hand, CoHA had an excellent antibacterial effect, as shown by *E. coli* evaluation, and the effect of TNT-CoHA powder was higher than that of Ti-CoHA powder. In summary, TNT-CoHA deposited electrochemically on the TNT substrates can be considered as a potential candidate for the application as an MRI contrast agent. This paper is a comparative study of how different electrode plates affect the magnetic and MRI image contrast of cobalt-substituted hydroxyapatite (CoHA) nanomaterials.

## 1. Introduction

Magnetic Resonance Imaging (MRI) is a safe diagnostic method. The use of a targeting material [1] or contrast agent enhances the contrast between normal tissue and pathological tissue [2,3]. Most of the contrast agents contain gadolinium ions (Gd^3+^), which may cause adverse reactions or complications in the body, such as nephrogenic systemic fibrosis (NSF) [4]. Therefore, the biocompatibility of the contrast agent is a very important issue. Currently, there is increasing demand for image contrast in the clinic. The magnetic dipole moment is induced in the magnetic nanoparticles under an applied magnetic field. When the water molecules diffuse to the rim of the induced dipole moment, the magnetic relaxation process of the water proton interferes, and the spin–spin relaxation time (T_2_) is shortened so that the semaphore intensity of the water molecules in T_2_-weighted (T_2_WI) imaging is reduced [5]. This phenomenon causes magnetic nanoparticles to be negatively contrasted in T_2_WI magnetic resonance imaging, reducing the strength of tissue semaphores in the image and increasing the contrast of the surrounding image [6]. Therefore, the contrast agent can increase the contrast by magnetic nanoparticles [7].

In addition, magnetic nanoparticles are widely used in cell labeling, magnetic drug delivery, cancer hyperthermia, and bone tissue engineering. Currently, iron oxide nanoparticles are commonly used in the application of contrast agents to effectively improve image contrast. However, there are reports that excessive iron oxide nanoparticles can cause accumulation of high doses of iron ions in localized areas of the body [8]. Hence, the development of non-toxic magnetic nanoparticles with a low dose and high image contrast is currently the main challenge. The chemical properties, crystal structure, and size and shape of magnetic nanoparticles will affect the magnetic properties of the contrast agent [9,10]. Previous literature has reported on ways to increase the strength of magnetic fields, for example, cobalt iron ferrite combined with carbon nanotubes used for an annealing treatment effectively enhanced the contrast of MRI images [11].

Among nanoparticles, cobalt is a trace element naturally present in the human body. Excessive cobalt ions are excreted by the kidneys, and there is no adverse reaction to the human body at a reasonable concentration [12]. Frank, S.J. et al. pointed out that cobalt chloride has a paramagnetic property and is a good candidate for use as contrast agent. Cobalt chloride can be combined with the drug acetylcysteine for use as an MRI marker to treat prostate cancer [13]. Cobalt ions can also promote angiogenesis and accelerate the repair of bone [14]. Previous research has applied cobalt ions in combination with hydroxyapatite for the treatment of bone defects. For example, Nenad Ignjatović et al. used the hydrothermal synthesis of cobalt-substituted hydroxyapatite (CoHA) to treat defects of the mandibular bone in rats. The results showed that there were no adverse reactions in the tissues around the CoHA after 24 weeks, and the bone creation was 1.2 times higher than that of pure hydroxyapatite (HA) [15]. Therefore, CoHA as a developing material does not have a negative impact on the human body, even if it is not metabolized quickly. In our previous work, we used electrochemical deposition to synthesize CoHA nanoparticles for bone repair. Preliminary analysis showed that CoHA is paramagnetic, allows for clear MRI image recognition, and is not toxic [16]. However, electrochemical deposition on different substrates can alter the properties of the coating. Titanium dioxide nanotube (TNT) substrates are a common choice for medical applications [17]. They have an ordered porous structure and a high surface area, which can be developed into a drug release and delivery platform [18,19].

This study investigated the effects of pure titanium and titanium dioxide nanotube substrates on the magnetic, magnetic resonance imaging, and biological behavior of cobalt-substituted hydroxyapatite. We hypothesized that: (1) using a porous TNT substrate to change the physicochemical properties of CoHA would improve the magnetic and MRI contrast, and (2) the substrates would not cause a significant difference to the biological behavior of CoHA.

## 2. Materials and Methods

### 2.1. Preparation of Titanium Dioxide Nanotubes (TNT)

A titanium plate with dimensions 8 × 4 cm (99.9%, Opetech materials, Hsinchu, Taiwan) was used as a titanium dioxide nanotube substrate. The titanium plate was sonicated for 10 min in acetone and another 10 min in ethanol and then washed with deionized water. The titanium metal was used as an anode and the cathode was a 304-stainless-steel plate in an electrolyte composition of 900 mL of ethylene glycol (Formosa Plastics, Taipei, Taiwan), 0.3 wt% ammonium fluoride (Acs 98.0%, Alfa Aesar, Lancashire, United Kingdom), and 2 vol% deionized water. The anodization was completed using a pulse power supply (GPR-50H10, GITEK ELECTRONICSCO, New Taipei, Taiwan). Electrolysis was run for 40 min while maintaining the temperature at 55 °C (Figure 1A). Finally, the TNT template was washed with ethanol ultrasonic shock for 20 min and then washed several times with deionized water.

### 2.2. Preparation of Cobalt-Substituted Hydroxyapatite

The cobalt-substituted hydroxyapatite was synthesized conforming to the method reported by Wei-Chun Lin et al. [16] with some minor modifications. The electrolytic solution was formulated with 25 mM ammonium dihydrogen phosphate (Shimada chemical works, Tokyo, Japan), 42 mM calcium nitrate (Shimada chemical works, Tokyo, Japan), and 7 mM cobalt chloride (Shimada chemical works, Tokyo, Japan). Pure titanium specimens or TNT were used as electrode plates to electrochemically synthesize cobalt-substituted hydroxyapatite (CoHA). The voltage was controlled at 5.5 volts for 20 min at 55 °C (Figure 1B). After the reaction, a plastic spoon was used to remove the powder from the electrode plate. The particles collected from the Ti and TNT plates were the samples used in the experiments (Ti-CoHA and TNT-CoHA).

### 2.3. Characterization

The morphology and particle size of TNT, Ti-CoHA, and TNT-CoHA were observed by a field emission scanning electron microscope (FE-SEM) (JSM-7610F, JEOL, Tokyo, Japan). In addition, the FE-SEM images were analyzed using Image-Pro Plus software (Media Cybernetics, Version 4 Maryland, MD, USA) to obtain the particle size of the powders. Surface element composition was detected for the CoHA by energy dispersive spectrometer (EDS, JEOL, Tokyo, Japan). The phase composition of the CoHA powders was determined by X-ray diffractometer (XRD) (Miniflex II, Rigaku, Tokyo, Japan) using CuKα radiation (λ = 1.54056 Å). The scanning conditions for this study were 4°/min and the scanning range was 10–70°. The crystal sizes of CoHA was calculated by the Scherrer equation:(1)$$\mathrm{Xs}=\frac{0.9\lambda }{\mathrm{FWHMcos}\theta }$$
where FWHM is the full width at half maximum of the diffraction peak and θ is the angle of the diffraction peak.

Fourier transform infrared (FTIR) spectra were recorded on a vertex80v spectrometer (Vertex80v, Bruker, Billerica, MA, USA) in the frequency range of 400 to 4000 cm^−1^, with 200 scans. Determination of the overall elemental composition in the Ti-CoHA and TNT-CoHA was determined by inductively coupled plasma optical emission spectrometry (ICP-OES) (Optima 8300, Perkin Elmer, Waltham, MA, USA). The magnetic properties of Ti-CoHA and TNT-CoHA were studied in an MPMS5 superconducting quantum interference devices (SQUID) magnetometer (MPMS5, Quantum Design, San Diego, CA, USA) in an applied magnetic field of −10,000 Oe to 10,000 Oe at 37 °C.

### 2.4. In Vitro MRI Experiments of CoHA and TNT-CoHA

MRI tests were performed on a 1.5 T MRI scanner (Signa Horizon LX, GE Healthcare, Chicago, IL, USA). A certain amount of CoHA and TNT-CoHA powders were dispersed in a gelatin (Sigma Aldrich, St. Louis, MO, USA) at different concentrations and then poured into 5 mL of distilled water. The sample and gelatin solutions were mixed thoroughly at 70 °C. The gelatin mixture was allowed to cool to room temperature. The T_2_-weighted images were acquired using spin echo imaging pulse sequence with the following parameters (matrix size = 256 × 256, field of view = 180 mm × 180 mm, slice thickness = 5 mm, echo time = 26 ms, repetition time = 100 ms, number of acquisitions = 2). We used a customized multiple-spin-echo pulse sequence to obtain T_2_ values. The sequence was developed to acquire eight spin echoes with an optional choice of echo spacing (ESP) from a minimum of 8.9 ms. Parameters of the multi-echo imaging protocol are as follows: Repetition Time (TR) = 4000 ms, Echo time (TE) = 8.9, 17.8, 26.6, 35.5, 44.4, 53.3, 62.2, 71.0 ms, slice thickness = 5mm, and number of acquisition = 2.

Color image processing was performed using Image J (1.51K, National Institutes of Health, Bethesda, Maryland, MD, USA). The contrast-enhancing efficacies of CoHA and commerce HA were determined by their relaxation coefficient (r_2_). The r_2_ was calculated using the following formula (equation) [20]:(2)$$\frac{1}{T_{2}}=\frac{1}{T_{2}^{0}}+r_{2}C$$
where $T_{2}$ is the observed relaxation time in the presence of CoHA, $T_{2}^{0}$ is the relaxation rate of pure gelatin, and C is the Co ion concentration.

### 2.5. Biocompatibility

#### 2.5.1. Cytotoxicity

The human osteosarcoma cell line (MG63) was maintained in a Dulbecco’s Modified Eagle Medium (DMEM) culture medium supplemented with 10% fetal bovine serum (FBS) (Biological Industries, Cromwell, CT, USA) and 1% penicillin/streptomycin. The cells were trained in a 5% CO_2_ incubator (310, Thermo Fisher Scientifc, Waltham, Massachusetts, MA, USA) at 37 °C. In this study, to investigate the effect of ions released by CoHA on cells. We used the CoHA extracts for testing, using 0.1 g of three CoHA powders, each immersed in 1 mL of deionized water and soaked for 7 days at 37 °C, followed by centrifugation, aspiration of the supernatant, and removal of the liquid using a freeze dryer. We then added 10 mL of DMEM to each sample and filtered the samples with a 0.22 μm sterile filtration apparatus as a culture medium for the cell tests. The MG63 cells were cultured in a 24-well cell culture plate (5 × 10^4^ cells/well) and incubated at 37 °C for 18 h. The extract with different samples (Ti-CoHA and TNT-CoHA) was added to the well and further incubated for 24 and 72 h. The in vitro biocompatibility of Ti-CoHA and TNT-CoHA was assessed by methyl thiazolyl tetrazolium (MTT) assays. The absorbance (O.D.) of the wavelength of 570 nm was read with an ELISA reader (Sunrise, Tecan, Männedorf, Switzerland). The biocompatibility was expressed as a percentage compared to that of the control tissue culture plate (TCP).

#### 2.5.2. Osteogenic Differentiation

Alizarin Red S (ARS) (Sigma Aldrich, St. Louis, MO, USA) is an orange-yellow needle-like crystal and an alizarin sulfonate sodium salt. Calcium salt was used to chelate the formation of an orange-red deposition complex to detect calcium deposition in cultured cells. Using 2% ARS and with a pH between 4.1 and 4.3, MG63 cells (1–10^4^ cells/well) were cultured with the extract for 3 and 7 days and then were washed 2–3 times with PBS and were fixed with 4% paraformaldehyde for 10 min. After removal, the cells were rinsed with deionized water once and ARS was added and allowed to react for 15 min at room temperature. Finally, we used deionized water to wash the cells 2–3 more times. In the quantitative analysis, the ARS that remained on the specimen after being washed with distilled water was dissolved in 0.2M NaOH/methanol (1:1) to measure the optical density at 620 nm [21]. 

#### 2.5.3. Antibacterial

In previous research it was determined that Co_3_O_4_ nanoparticles attach to bacterial surfaces through electrostatic forces and van der Waals forces. The surface of Co_3_O_4_ is covered by hydroxyl groups in an aqueous environment. Therefore, the surface is positively charged in an acidic environment and negatively charged in an alkaline environment [22]. The surface of Escherichia (*E. coli*) is negatively charged at pH 6.5 [23]. The previous results show that the contact between *E. coli* and Co_3_O_4_ becomes active by electrostatic interaction, which contributes to the improvement of the antibacterial activity of Co_3_O_4_. Therefore, we investigated CoHA against *E. coli*, as a model of Gram-negative bacteria, by the colony plate-count method in order to quantify the bacterial effect of our system [24]. The *E. coli* were prepared from a fresh brain–heart infusion (BHI, Becton Drive, Franklin Lakes, NJ, USA) and incubated at 37 °C for 24 h. The BHI containing *E. coli* was diluted to 10^−3^ of its original concentration; 1 mL of the bacteria liquid was extracted and added to a fixed weight (6 mg) of Ti-CoHA and TNT-CoHA and then placed into a centrifuge tube and cultivated for 18 h. After the samples were removed, a 100-μL bacterial solution was extracted and applied on the BHI agar (Becton, Dickinson and Company, Houston, TX, USA) petri dish before being cultured for 24 h at 37 °C. Finally, the colonies were counted, and the results were expressed as percentage reduction rates using bacteria number = [α × 10^5^], where α is the number of bacterial colonies.

#### 2.5.4. In Vitro Biodegradation

One tenth of a gram each of Ti-CoHA and TNT-CoHA were immersed in 10 mL of phosphate buffered saline (PBS), placed in a constant temperature water bath (37 °C) for four weeks, and the pH was measured daily. The PBS composition was 0.2 g KCL, 0.2 g KH_2_PO_4_, 8 g NaCl, and 2.16 g Na_2_HPO_4_.

### 2.6. Statistical Analysis

All data are from the average of three repetitions ± standard deviation. Data were calculated using JMP14 software (JMP^®^14.1.0, Cary, NC, USA). One-way ANOVA was used to examine the differences between groups using the Tukey HSD multiple comparisons. A value of *p* < 0.05 was considered to be significant. 

## 3. Results and Discussion

### 3.1. Characterization of Titanium Dioxide Nanotubes (TNT)

The pure titanium test piece was anodized to obtain a flat TNT (Figure 2B). The SEM image shows that the outer diameter and the wall thickness of the nanotubes are 34.6 ± 2.9 nm and 10.1 ± 2.0 nm, respectively. The TNT that have been reported on so far have had an outer diameter of about 4 to 400 nm. According to reports, the smaller the diameter of TNT, the better is the obtained conductivity [17]. The TNT surface contained titanium, oxygen, and fluorine (Figure 2D), as determined by surface elemental analysis.

### 3.2. Characterization of Cobalt-Substituted Hydroxyapatite (CoHA)

The appearance of the Ti-CoHA and TNT-CoHA powders was observed to be spherical by SEM (Figure 3A–D). FE-SEM images were analyzed using Image-Pro Plus software. The particle sizes of Ti-CoHA and TNT-CoHA were 418.6 ± 8.0 nm and 127.5 ± 2.9 nm, respectively. An increase in the surface area of the TNT plate resulted in a reduction in particle size. It is relatively simple to control the synthesis of spherical magnetic nanoparticles and their particles are evenly distributed [25]; this makes them a useful material for MRIs. Moreover, the spherical nanoparticle structure of the hydroxyapatite is more favorable for the growth of osteoblasts than a rod-like structure [26]. The cobalt ion content of Ti-CoHA and TNT-CoHA surfaces was 3.34% and 2.86%, respectively. The overall chemical composition by ICP-OES analysis showed that the cobalt ion content of Ti-CoHA and TNT-CoHA were 14.0% and 19.2%, respectively (Table 1). The results show that most of the cobalt oxide in Ti-CoHA was dispersed on the surface, and that of TNT-CoHA was concentrated inside. The calcium to phosphorus ratio of TNT-CoHA was less than 1.67, which indicates a calcium-deficient hydroxyapatite. Previous studies have indicated that calcium-deficient hydroxyapatites have a higher degradability [27]. They are easily degraded in the body to provide absorption of surrounding tissues. The crystal structures of the two CoHAs were analyzed by XRD (Figure 4A). We found hydroxyapatite diffraction peaks (JCPDS.No-09-0432) at 25.9°, 32.0°, and 40.6° [28] and cobalt oxide diffraction peaks (JCPDS. No-42-1467) at 18.6° and 37.6° [24]. TNT-CoHA showed diffraction peaks of TiO_2_ (ICSD. No-024080) at 28.9° and 30.0° [29]. TNT-CoHA was affected by TNT and reduced the diffraction peak intensity of the HA. As more calcium ions were replaced by cobalt ions, the strength of the hydroxyapatite was lowered [30,31]. The change in the internal functional groups of the Ti-CoHA and TNT-CoHA was analyzed by FTIR (Figure 4B). The absorption peaks observed for all samples were essentially similar. The absorption peaks at wave numbers of 575 cm^−1^ and 1053 cm^−1^ represent ${\mathrm{PO}}_{4}^{3-}$, that at 1412 cm^−1^ represents ${\mathrm{CO}}_{3}^{2-}$, while those at 1639 cm^−1^ and 3438 cm^−1^ represent −OH [28]. However, it was found in the −OH group that TNT-CoHA was offset compared to Ti-CoHA. It appears that the combination of TNT and CoHA results in a peak shift of −OH.

### 3.3. Magnetic Analysis

Being superparamagnetic is an important condition for nanoparticles to possess. Superparamagnetic nanoparticles can be applied to stem cell therapy, drug delivery, and contrast agents. Figure 5A shows that both Ti-CoHA and TNT-CoHA are paramagnetic materials. The hysteresis loop is fine and narrow, indicating the TNT-CoHA is a soft-magnetic material [32]. Soft-magnetic materials will quickly lose their magnetic properties when the applied magnetic field is removed. It is possible to reduce the nanoparticles from being agglomerated by the magnetic field to cause embolization in the blood vessels. The coercivity (*Hc*), saturation magnetization (*Ms*), and remnant magnetization (*Mr*) of Ti-CoHA and TNT-CoHA are summarized as Table 1. The squareness ratio (*Mr*/*Ms*) is an important characteristic parameter for magnetic material applications. The literature indicates that the squareness ratio of a material will be reduced due to the decrease in Co content [33]. Therefore, Ti-CoHA is less valuable than TNT-CoHA. At the same time, the ratio of *Mr*/*Ms* of the two samples is less than 0.5, indicating that the particles interact through magnetostatic interactions [34]. It is worth noting that the *Ms* and *Mr* of TNT-CoHA are 12.6 times and 13.5 times larger than those of Ti-CoHA, respectively. Magnetic properties depend on the magnetization direction, morphology, and crystallinity of the material [35]. V. Sarath Chandra et al. reported that with the hydrothermal–microwave technique, the synthesized CoHA showed a reduction in particle size due to an increase in the strength of the magnetic field (207–274 nm) [36]. Those results are similar to other results reported in the literature, i.e., smaller particles have higher magnetism. However, many magnetic nanoparticles show a magnetic field strength that increases with the particle size (5–25 nm) [20,37]. Since hydroxyapatite is a diamagnetic material, it imparts superparamagnetism by the introduction of cobalt ions. Therefore, we calculated the crystal size by the position of cobalt oxide (18.6°) in the XRD pattern. The cobalt oxide crystal sizes of CoHA, calculated by the Scherrer equation [38], were approximately 2.7 nm for Ti-CoHA and 3.3 nm for TNT-CoHA. In addition, we compiled details from the literature of the effects of the crystal size of the magnetic nanoparticles on the saturation magnetic field (Figure 5B). The results show that the crystal size of all magnetic nanoparticles is proportional to the magnetic field strength. In summary, TNT-CoHA has a smaller particle size and a larger cobalt oxide crystal size relative to Ti-CoHA, thus reducing the distance between the cobalt oxides and enhancing the magnetostatic interaction between the particles. This explains why the magnetic field strength of TNT-CoHA is greater than that of Ti-CoHA. Currently, the use of large-sized magnetic nanoparticles can effectively increase the *Ms*. However, the increasing particle size may be detrimental to pharmacokinetics and biodistribution, so the development of small magnetic nanoparticles with high magnetic saturation is the main target [39]. We used CoHA magnetic nanoparticles deposited on TNT plates to meet this need.

### 3.4. In Vitro MRI Examination

The magnetic dipole moment is induced in superparamagnetic nanoparticles under an applied magnetic field. This change results in the magnetic nanoparticles being a negative contrast in T_2_WI magnetic resonance imaging, reducing the intensity of tissue signals in the image and increasing the contrast of the surrounding image [6]. Commercialized HA showed no change in T_2_WI imaging with increasing concentration, but Ti-CoHA and TNT-CoHA improved the image contrast (Figure 6A). The CoHA concentration corresponds to the change in the T_2_WI MRI (Figure 6B). The results showed that both samples had linear growth, and the slope of TNT-CoHA was greater than that of Ti-CoHA. This signifies that TNT-CoHA has better image sensitivity. The spin–spin relaxivity R_2_ (R_2_ = 1/T_2_) is usually used to indicate the paramagnetic effect. The higher the R_2_, the shorter the T_2_ relaxation time, the darker the tissue image, and the larger the contrast effect compared with the tissue with lower R_2_. In addition, the R_2_ value of TNT-CoHA (R_2_ = 46) was superior to that of Ti-CoHA (R_2_ = 33) at a cobalt ion concentration of 0.12 Mm (Figure 6B). Previous studies have shown that *Ms* is proportional to the value of R_2_ [9,40,41]. Therefore, TNT-CoHA has a good T_2_ image effect. The relaxivity coefficient (r_2_) obtained from the gradient of the R_2_ curve and the magnetic nanoparticle molar concentration is a standardized contrast intensity index [5]. The higher the r_2_ value, the lower the concentration of nanoparticles required in the same T_2_ image. This avoids excessive metal ion accumulation. The calculated relaxivity coefficient of the CoHA magnetic nanoparticles was higher for TNT-CoHA (340.3 mM^−1^s^−1^) than Ti-CoHA (211.7 mM^−1^s^−1^). However, there are many factors affecting the relaxation efficiency coefficient such as particle size, crystal size, surface effect, and saturation magnetic field strength [9]. The SEM image shows that the particle size of Ti-CoHA is larger than that of TNT-CoHA. The calculated surface area of a single particle of Ti-CoHA was 10.8 times higher than that of TNT-CoHA. Therefore, when the weights of the particles are the same, the surface effect of TNT-CoHA is higher than that of Ti-CoHA. In Figure 6A, TNT-CoHA has a good negative contrast when the powder weight is the same. This results in more water molecules binding to the TNT-CoHA surface to reduce the water signal in T_2_WI imaging and thus increases the image contrast. Another plausible reason is attributed to the fact that the *Ms* of TNT-CoHA is higher than that of Ti-CoHA. We put together a comparison of the magnetic nanoparticle relaxivity coefficients reported in the literature (Figure 6C). As described above, the relaxivity coefficient of the nanoparticle increases with *Ms*. It is worth noting that although Ti-CoHA and TNT-CoHA have lower *Ms* values than other magnetic nanoparticles, their relaxivity coefficients are relatively high. Since the surface of the CoHA is rich in many OH groups, it can easily bind to water molecules. These bonds may result in a relatively high relaxivity coefficient of Ti-CoHA and TNT-CoHA. In addition, Adamiano et al. reported the comparison of iron-substituted hydroxyapatite (FeHA) with commercial iron nanoparticles Endorem^®^ NP (Guerbet, France) as an MRI contrast agent in the liver [8]. The results showed that the r_2_ values of FeHA and Endorem^®^ NP were 165 mM^−1^s^−1^ and 103 mM^−1^s^−1^, respectively, and both showed a good contrast. This study and the aforementioned FeHA both used ion-substituted HA. Since the r_2_ values are higher in this study than in [18], this indicates that Ti-CoHA and TNT-CoHA have considerable potential as MRI T_2_WI contrast agents. Furthermore, the magnetic nanoparticle size clinically approved for use in contrast agents is 10–300 nm [42]. The TNT-CoHA size is 127.5 nm, this is within the approved range. The literature indicates that nanoparticles with a diameter greater than 100 nm are prone to cellular uptake [5]. Nanoparticles with a diameter in the range of 1–30 nm can shuttle between cells and blood vessels [43]. In the present study, the purpose of TNT-CoHA is to target the location of cells and tissues so that cells do not arbitrarily shuttle after phagocytosis and degrade and absorb in the body.

### 3.5. Biocompatibility

Cytotoxicity testing was performed using the MG63 cell line, and the tissue culture plate (TCP) was used as a control group. MTT results showed no significant differences in initial cell attachment (Figure 7A). The survival rate of TNT-CoHA cells in the late stage of cell culture was significantly higher than that of the TCP (*p* ˂ 0.05). The quantitative results of calcium deposition by Alizarin Red S (ARS) staining are shown in Figure 7B. Calcium deposition of Ti-CoHA and TNT-CoHA was significantly higher than that of TCP (*p* ˂ 0.05) after 3 and 7 days of cell culture. The results showed that the two groups of CoHA were not toxic and TNT-CoHA contributed to cell growth. In addition, bacterial infections around the wound are an important challenge for bone repair materials. Previous studies have reported that Co_3_O_4_ effectively inhibits the growth of *E. coli* and can be used as an antibacterial drug [24]. Accordingly, we used *E. coli* to evaluate the antibacterial properties of CoHA by colony counting. The number of colonies in the TNT-CoHA was significantly lower than that of the other groups after 24 h of culture (Figure 7C). Quantitative results showed that the two groups of samples had the effect of inhibiting *E. coli*, and TNT-CoHA was the best (*p* ˂ 0.05). Residues of the contrast agent and target in the body may have a negative impact on the human body. Therefore, CoHA was immersed in PBS to simulate changes in the pH of the human body (Figure 7D). The results showed that Ti-CoHA reached equilibrium after 5 days of immersion, while TNT-CoHA continued to decrease. It is well known that cations are acidic and anions are alkaline. TNT-CoHA leads to a lower pH due to late cation release. This indicates that TNT-CoHA degrades faster and is easily decomposed, which is consistent with the Ca/p ratio. Taken together, the two substrates differ in the biological behavior of CoHA. The smaller size of TNT-CoHA has a higher surface area that can react with cells and bacteria while accelerating the effectiveness of particle degradation.

## 4. Conclusions

In this study, we synthesized Ti-CoHA and TNT-CoHA magnetic nanoparticles by electrochemical deposition using two different substrates. It was found that the crystal size of the cobalt oxide of the nanoparticle strongly influenced the degree of magnetization and the relaxivity coefficient. Although the particle size of TNT-CoHA is smaller than that of Ti-CoHA, the internal cobalt oxide of TNT-CoHA has a larger crystal size than Ti-CoHA, thus, TNT-CoHA has a saturation magnetic field strength of up 12.6 times that of Ti-CoHA. The T_2_ image shows that TNT-CoHA has good image contrast at low concentrations, avoiding the accumulation of high doses of magnetic nanoparticles in the body. The relaxivity coefficient of the sample showed that TNT-CoHA (340.3 mM^−1^s^−1^) was higher than that of Ti-CoHA (211.7 mM^−1^s^−1^), and both were higher than that of the commercial iron nanoparticles (103.0 mM^−1^s^−1^). In addition, Ti-CoHA and TNT-CoHA are not cytotoxic. In short, the use of titanium dioxide nanotubes as a substrate effectively improves the saturation magnetic field strength and magnetic resonance contrast of CoHA and has good biocompatibility and antibacterial properties. These results indicate that TNT-CoHA might be effectively used as an MRI contrast agent or targeting material in the future and may be helpful for the development of medical imaging materials.

## Figures and Tables

**Figure 1 nanomaterials-09-00200-f001:**
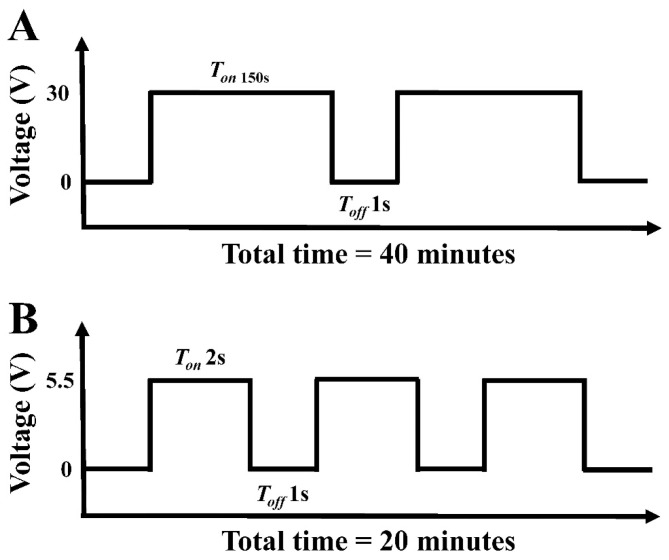
Schematic diagram of the pulse current used in the (**A**) anodization of titanium dioxide nanotubes and (**B**) electrodeposition of cobalt-substituted hydroxyapatite.

**Figure 2 nanomaterials-09-00200-f002:**
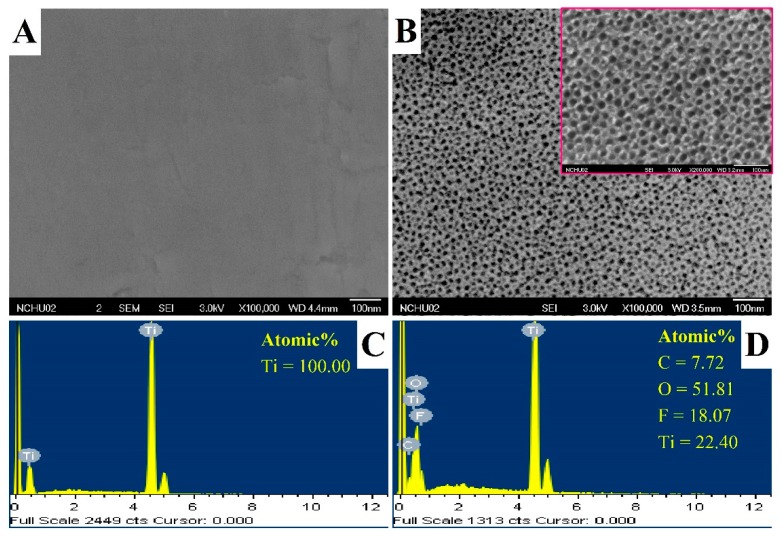
The surface morphology and composition of the (**A**,**C**) pure titanium and (**B**,**D**) titanium dioxide nanotube (TNT) as observed by FE-SEM and EDS, respectively. The red frame is a magnified image of the TNT.

**Figure 3 nanomaterials-09-00200-f003:**
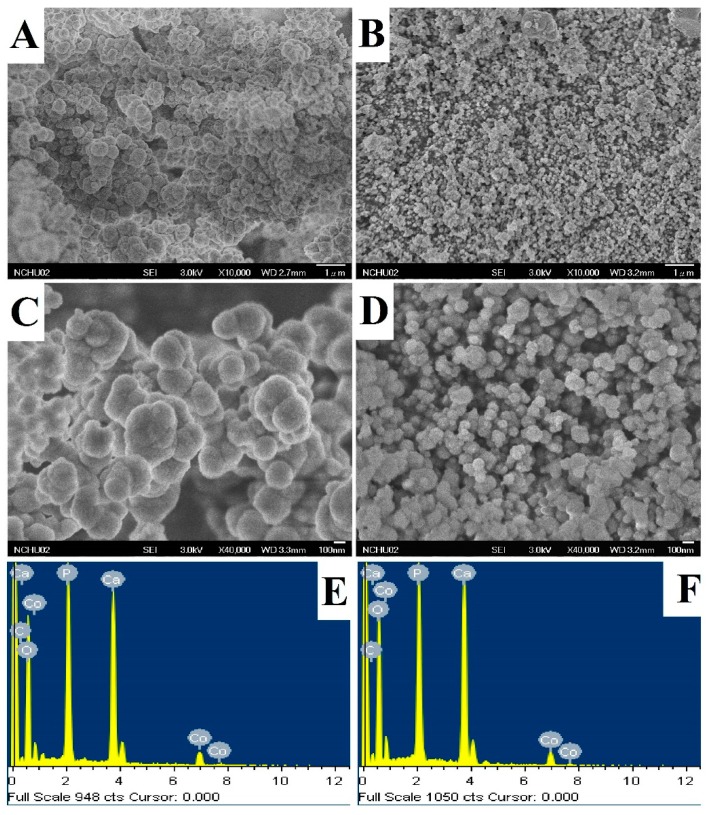
The surface morphology and composition of the (**A**,**C**,**E**) Ti-CoHA and (**B**,**D**,**F**) TNT-CoHA as observed by FE-SEM and EDS. CoHA stands for cobalt-substituted hydroxyapatite.

**Figure 4 nanomaterials-09-00200-f004:**
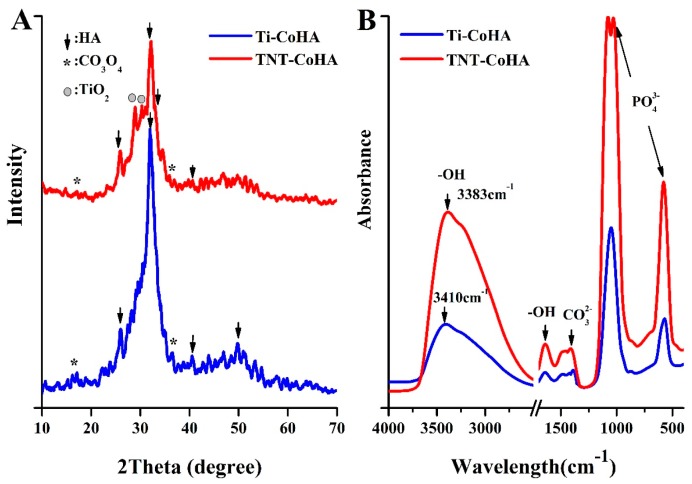
The crystal structure and chemical composition of Ti-CoHA and TNT-CoHA: (**A**) XRD patterns, (**B**) FTIR spectra.

**Figure 5 nanomaterials-09-00200-f005:**
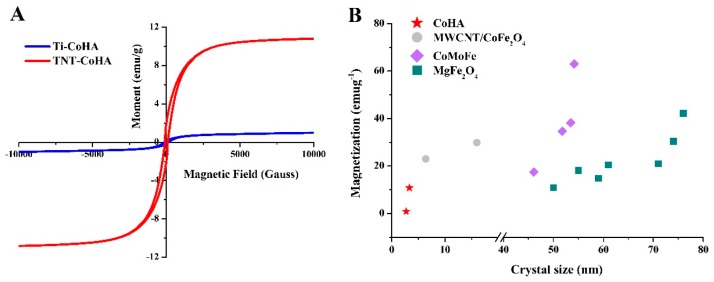
(**A**) Mass magnetization of CoHA and TNT-CoHA by a superconducting quantum interference device (SQUID). (**B**) Magnetic nanoparticle crystal size effects on magnetic behavior by XRD. Other magnetic nanoparticles data taken from MWCNT/CoFe_2_O [11], CoMoFe [33], and MgFe_2_O_4_ [34].

**Figure 6 nanomaterials-09-00200-f006:**
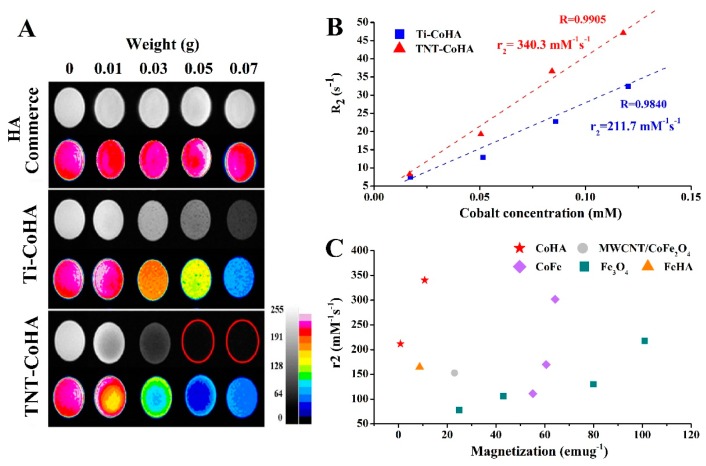
(**A**) T_2_-weighted MRI images of hydroxyapatite (HA) (commerce), CoHA, and TNT-CoHA suspended in gelatin at different concentrations. (**B**) The T_2_ relaxation rate R_2_ (1/T_2_) against cobalt concentration of Co-HA. (**C**) Magnetic nanoparticle magnetization effects on relaxivity coefficient r_2_. Other magnetic nanoparticles data taken from MWCNT/CoFe_2_O [11], CoFe [20], Fe_3_O_4_ [44], and FeHA [8].

**Figure 7 nanomaterials-09-00200-f007:**
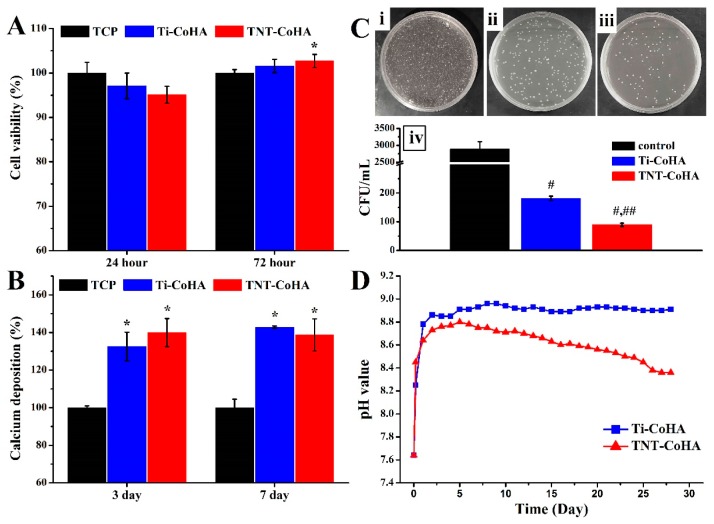
(**A**) Cytotoxicity test using MG63 cells. (**B**) Quantitative analysis of matrix deposition mineralization was performed at 3 and 7 days. (**C**) Images of the antibacterial results for relatively bacterial colonies on agar plates: (i) control (brain–heart infusion (BHI)), (ii) Ti-CoHA, (iii) TNT-CoHA, (iv) antibacterial activity of the analyzed sample on *E. coli*. (**D**) The pH values of different Ti-CoHA and TNT-CoHA materials soaked in PBS solution. (*: Significantly higher than the control group (tissue culture plate (TCP)), #: Significantly lower than the control group (BHI), ##: Significantly lower than other groups, *p* < 0.05, mean ± SD, *n* = 4).

**Table 1 nanomaterials-09-00200-t001:** The particles size, composition, and magnetic analysis of Ti-CoHA and TNT-CoHA.

Sample	Particle Size (nm)	Surface Composition *	Overall Composition **		Magnetic Field Strength			
		Co+Ca/P (%)	Co+Ca/P (%)	X_Co_ (%)	*Hc* (Oe)	*Ms* (emu/g)	*Mr* (emu/g)	Squareness Ratio (*Mr*/*Ms*)
Ti-CoHA	418.6 ± 8.0	1.67	1.67	14.0	133.98	0.86	0.15	0.17
TNT-CoHA	127.5 ± 2.9	1.43	1.62	19.2	89.68	10.82	2.02	0.19

*: by energy dispersive spectrometer (EDS). **: by inductively coupled plasma optical emission spectrometer (ICP-OES).

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
