# Peer review of "The Effect of Electrode Topography on the Magnetic Properties and MRI Application of Electrochemically-Deposited, Synthesized, Cobalt-Substituted Hydroxyapatite"

_nanomaterials, 2019, doi:10.3390/nano9020200_

Reviewer 1 Report

The paper submitted by C. M. Tang describes the synthesis and the biotoxicity evaluation of cobalt-substituted hydroxyapatite nanoparticles deposited on titanium and on titanium dioxide nanotubes.

The subject is of interest, the application of this type of NP being wide. Some results presented in the submitted paper are well documented and explained but there are inconsistencies that must be addressed by the authors:

1.     In the title of the paper the effect of the electrode topography is mentioned but no details regarding this subject are presented in the experimental part or in results and discussion part. The reference 21 is not accessible; therefore I suggest adding some relevant information regarding the electrode topography influence and the process optimization. The equipments used in this study must also be mentioned.

2.     More explanations are needed for Fig 1 in results and discussions.

3.     There are several phrases incompletes and the entire experimental part must be reconsidered due to the fact that it cannot be understood. I suggest that the entire manuscript should be revised from language/grammar point of view.

4.     What is the meaning of “developers” us for cobalt chloride?

5.     Please replace pharmacy with drug (for acetylcysteine).

6.     In experimental par for 2.5.3. please add antibacterial effect.

Author Response

Response to Reviewer 1 Comments

Q1. In the title of the paper the effect of the electrode topography is mentioned but no details regarding this subject are presented in the experimental part or in results and discussion part. The reference 21 is not accessible; therefore I suggest adding some relevant information regarding the electrode topography influence and the process optimization. The equipments used in this study must also be mentioned.

A1: Thanks for your suggestion.  Currently, the study of electrochemical deposition is mainly focused on the electrolysis parameters changing [1,2]and composition of the electrolytic solution[3].  In our study, the novelty is to explore the effect of the surface topography of the electrode plate on the product. Therefore, the main body of research is still on CoHA.  Section 3.2 of the article, we also mentioned that “The Ti-CoHA and TNT-CoHA particles were 418.6 ± 8.0 nm and 127.5 ± 2.9 nm, respectively. An increase in the surface area of the TNT plate is shown to result in a reduction in particle size”.  We are updated the reference 21 and the source of the equipment used in this study was revised on article

 Q2. More explanations are needed for Fig 1 in results and discussions.

A2: Fig 1 is a schematic diagram of the power supply conditions.  Mainly to make it easier for readers to understand the electrolysis parameters on the material method.

 Q3. There are several phrases incompletes and the entire experimental part must be reconsidered due to the fact that it cannot be understood. I suggest that the entire manuscript should be revised from language/grammar point of view.

A3: Thanks for your suggestion.  We will to English editing of manuscript.

 Q4. What is the meaning of “developers” us for cobalt chloride?

A4: Thanks for reviewer correction.  The word "developer" is corrected to "contrast agent" on article.

 Q5. Please replace pharmacy with drug (for acetylcysteine).

A5: Thanks for reviewer correction.  The word "pharmacy" is corrected to "acetylcysteine drug” on article.

 Q6. In experimental par for 2.5.3. please add antibacterial effect.

A6: Thanks for your suggestion. We added the effect of cobalt oxide on antibacterial in the material method.  In section of 2.5.3, the under description was added.  “In previous literature, Co3O4nanoparticles can be attached to bacterial surfaces by through electrostatic force and van der Waals force. The surface of Co3O4is covered by hydroxyl groups in an aqueous environment. Therefore, the surface is positively charged in an acidic environment and negatively charged in an alkaline environment [4]. The surface of Escherichia (E. coli) is negatively charged at pH 6.5 [5]. The results show that the contact between E. coli and Co3O4becomes active by electrostatic interaction, which contributes to the improvement of the antibacterial activity of Co3O4.

References

1.         Parcharoen, Y.; Kajitvichyanukul, P.; Sirivisoot, S.; Termsuksawad, P. Hydroxyapatite electrodeposition on anodized titanium nanotubes for orthopedic applications. Applied Surface Science 2014311, 54-61.

2.         Li'nan, J.; Chenghao, L.; Naibao, H.; Feng, D.; Lixia, W. Formation and characterization of hydroxyapatite coating prepared by pulsed electrochemical deposition. Rare Metal Materials and Engineering 201544, 592-598.

3.         Suchanek, K.; Bartkowiak, A.; Gdowik, A.; Perzanowski, M.; Kac, S.; Szaraniec, B.; Suchanek, M.; Marszalek, M. Crystalline hydroxyapatite coatings synthesized under hydrothermal conditions on modified titanium substrates. Mater Sci Eng C Mater Biol Appl 201551, 57-63.

4.         Wang, N.; Hsu, C.; Zhu, L.; Tseng, S.; Hsu, J.P. Influence of metal oxide nanoparticles concentration on their zeta potential. J Colloid Interface Sci 2013407, 22-28.

5.         Jiang, W.; Mashayekhi, H.; Xing, B. Bacterial toxicity comparison between nano- and micro-scaled oxide particles. Environ Pollut 2009157, 1619-1625.

Reviewer 2 Report

The submitted text describes synthesis and characterisation of the potential MRI contrasting magnetic nanoparticles Ti-CoHA and TNT-CoHA. While the overall idea for this engineering and research effort is good, the presentation of it in the paper require substantial improvement. First of all, extensive language correction is necessary. Practically in entire text there are many syntax and grammar errors.  I would suggests correction by native speaker or specialised correction service.

Secondly, due to placing results and discussion in one section, reader has sometime impression of chaos once trying to follow author's argumentation. I would suggest more structuring of the text within subsections, which separates better description of the results from (the following) discussion.

The more detailed comments:

- what do you mean,  using word "developer" ?  Especially the term "MRI developer" (in Conclusion) has for sure completely different meaning than you (probably) have in mind writing it.

Subsection 2.4

- How do you obtained T2 (and then r2) values from the MRI data with only one TE, as it is indicated in  ? Please describe it in more details, as it is not clear how you obtained results presented in Fig. 6B.

Subsection 3.4:

-reducing the intensity of nanoparticle-loaded tissue in T2-weighted image does not mean making the surrounding area brigther,

- it is not true that T2 in TNT-CoHA is higher than in Ti-Co-HA. It is 1/T2.

Author Response

Response to Reviewer 2 Comments

Secondly, due to placing results and discussion in one section, reader has sometime impression of chaos once trying to follow author's argumentation. I would suggest more structuring of the text within subsections, which separates better description of the results from (the following) discussion. 

Thanks for reviewer suggestion.  We are re-edit this article so that readers can understand more clearly.

 The more detailed comments:

Q1. What do you mean, using word "developer"?  Especially the term "MRI 

Developer" (in Conclusion) has for sure completely different meaning than you (probably) have in mind writing it. 

A1: Thanks for reviewer correction.  The word "developer" is corrected to "contrast agent" on article.

 Subsection 2.4

Q2. How do you obtained T2 (and then r2) values from the MRI data with only one TE, as it is indicated in? Please describe it in more details, as it is not clear how you obtained results presented in Fig. 6B. 

A2: First, the sample is analyzed by MRI according to the parameter setting(matrix size = 256 x 256, field of view = 180 mm x 180 mm, slice thickness = 5 mm, echo time = 26 ms, repetition time = 100 ms, number of acquisitions = 2), and automatic calculation of T2through computer.  Convert T2to R(1 / T2) value (For Fig. 6B 1/T2vs. concentration).  Finally, the relaxivity coefficient (r2) obtained from the gradient of the R2curve and the molar concentration of the magnetic nanoparticles [1].  The r2is calculated by the following formula(eq 1):[2]

   (eq 1)

where  is the observed relaxation time in the presence of CoHA,  is the relaxation rate of pure gelatin and  is the Co ion concentration.

 Subsection 3.4: 

Q3. Reducing the intensity of nanoparticle-loaded tissue in T2-weighted image does not mean making the surrounding area brigther, it is not true that T2in TNT-CoHA is higher than in Ti-Co-HA. It is 1/T2

A3: Thanks for reviewercorrection.  The correction was added to the article.  This change results in magnetic nanoparticle as a negative contrast in T2WI magnetic resonance imaging, reducing the intensity of tissue signals in the image and increase the contrast of surrounding images [3].  In addition, the T2value of TNT-CoHA (46) was superior to that of Ti-CoHA (33) at a cobalt ion concentration of 0.12 mM.

 References

1.         Jun, Y.W.; Lee, J.H.; Cheon, J. Chemical design of nanoparticle probes for high-performance magnetic resonance imaging. Angew Chem Int Ed Engl 2008,47, 5122-5135.

2.         Joshi, H.M.; Lin, Y.P.; Aslam, M.; Prasad, P.V.; Schultz-Sikma, E.A.; Edelman, R.; Meade, T.; Dravid, V.P. Effects of shape and size of cobalt ferrite nanostructures on their mri contrast and thermal activation. J. Phys. Chem. C 2009, 17761–17767.

3.         Kalambur, V.S.; Han, B.; Hammer, B.E.; Shield, T.W.; Bischof, J.C. In vitrocharacterization of movement, heating and visualization of magnetic nanoparticles for biomedical applications. Nanotechnology 200516, 1221-1233.

Reviewer 3 Report

This works describes a development of new contrast agents, cobalt based nanoparticles with Ti and Ti nanotubes. The new agents are analyzed with several analytical techniques including SEM/EDS, XRD, FTIR, and SQUID. The effects on MRI contrast are studied, and biological assays are also carried out. They conclude that their new contrast agents, especially Co-hydroxyapatite nanoparticles on Ti nanotubes, are very effective in enhancing MRI contrasts and have low biological side effects than the popularly utilized Gd ion based commercial agent. However, there are many ambiguous statements in this manuscript. Not only its English grammar, but English styles/presentations need to be improved greatly.   
In Abstract:
1) In the other hand, the development of magnetic nanoparticles with a low dose, high image recognition and non-toxicity is currently a major challenge.   

 "high image recognition" probably "high image contrast."
2) The results show that the particle sizes of Ti- CoHA and TNT-CoHA are 418.6 nm and 127.5 nm, respectively. Although the particle size of TNT-CoHA is smaller than Ti-CoHA. However, the grain size of cobalt oxide in TNT-CoHA is larger than cobalt oxide in Ti-CoHA.
What does this mean? They should state which results give the larger (>100nm) particle size, and what results give the grain sizes to be different. In fact, particle size and grain size need to be defined.
Introduction
3) I suggest to use the term “contrast agent” instead of “developer” throughout the document.   
4)  Introduction should perhaps start with the mechanism on how the nanoparticles enhance the MRI contrasts.
5) "Frank SJ et al. pointed out that cobalt chloride has a paramagnetic property and is a good candidate for developers." This sentence should be followed with a reference to this work.
6) "Combined with Acetylcysteine pharmacy can be applied to MRI marker for prostate cancer treatment[14]." There is no subject in this sentence, or is "pharmacy" the subject indeed?
7) Is hydroxyapatite anniversary applicable to all organs or mostly for bone tissues?   

2. MATERIALS AND METHODS 
8) "fig.1"  should be "Fig.1"

9) The crystallite sizes of CoHA, calculated by the Scherrer equation [22]:
So far, the authors used “particle size,” “grain size,” and “crystal size,” in the manuscript. They need to strictly define what these terms are throughout the manuscript. 
10) Strictly speaking, they should keep the same style in providing the manufacture/vendor information. Some are defined with country only, and the others are defined with City/Country.
3. RESULTS AND DISCUSSION  

11) fig. 2  should be Fig. 2.
12) Along with the EDS spectra in Fig. 2, compositions obtained by EDS should be presented. The readers want to know the compositions of the impurities, such as F and C, as well as the Ti to O ratios for both samples.   
13) "The Ti-CoHA and TNT-CoHA particles were 418.6 Å} 8.0 nm and 127.5 Å} 2.9 nm, respectively." Should mention the details of how these particle sizes were obtained. I think, you can see the smaller "grains" consisting of these "particles" by looking at the images at higher resolutions.
14) Need to discus why the two sets of data (EDS vs. IPC) on Co concentration are different here more quantitatively. What was the electron energy used in obtaining the EDS spectra which affects the depth of information (how deep were the X-rays collected by EDS to yield the Co concentration) under this condition. This will in tern, define what the authors claim to be "surface."  
15) On Fig. 4, the entire XRD spectra (10-70degrees) should be given in addition to the narrow range scans given in Fig. 4A. At least in the ranges to show all the peaks mentioned in the body.    
16) On Fig. 4 B, blue line spectrum (Ti-CoHA) is lower than Red line spectrum. Is this due to the difference in the amount of the powders photons traverse? If so, can you normalize them by using for example, 2 PO4 lines, which would make the argument about the difference in the OH groups meaningless. 
17) The cobalt oxide crystallite sizes of CoHA, calculated by the Scherrer equation[22], were about 2.7 nm for Ti-CoHA and 3.3 nm for TNT-CoHA, respectively.   

No data (spectra shown for these XRD peaks. With the peaks shown in Fig. 4 A for another peak, the uncertainties in the FWHM may not be small. How confident are the authors on this data (3.3nm vs. 2.7nm)?  Please give the estimated experimental error values.
18) Fig. 5B; The results show that the crystal size of all magnetic nanoparticles is proportional to the magnetic field strength.
I do not get this statement at all from Fig. 5 B.

19) In summary, TNT-CoHA has a smaller particle size and a larger cobalt oxide size relative to Ti-CoHA This statement does not make sense either without the strict definitions of the particle/grain/crystal sizes.   
3.4 In vitro MRI examination  

20) The relaxivity coefficient (r2) obtained from the gradient of the R2 curve and the magnetic nanoparticle molar concentration is a standardized contrast intensity index. [42]. The higher the r2 value, the lower the concentration of nanoparticle required in the same T2 image.
Are lower case r2s simple typing mistakes of R2, or does lower case r2 mean something else?

21)Calculating the overall surface area of the particles indicated that Ti-CoHA was 10.8 times higher than TNT-CoHA. That is to say, when the concentration of the particles is the same, the surface effect of TNT-CoHA is higher than that of Ti-CoHA.

Explain how this conclusion was obtained? Can you back it up by the experiments utilizing Surface Area Analyzer (such as Micromeritics ASAP 2020)? Are the author simply taking the average size of the particles of the two samples and comparing the surface areas of the same volume? Is assuming the concentrations of the two agents to be equal a reasonable assumption? Why not use the same weights for the two? Most importantly, can you quantitatively explain the difference shown in Fig. 6 with the increase in the surface area alone (I do recognize that the authors mention that many factors affect inn the results.)?
22) The literature indicates that nanoparticles with a diameter greater than 100 nm are prone to cellular uptake [47].
Are you stating that nanoparticles less than 100 nm in diameter are not likely to be uptaken by cells? Or, stating that they are larger particles are more likely to be uptaken that the nanoparticles with less than 100 nm dia.? Can you quantify the difference in the rates of uptake?
23) The literature indicates that nanoparticles with a diameter greater than 100 nm are prone to cellular uptake [47].
Are you stating that nanoparticles less than 100 nm in diameter are not likely to be uptaken by cells? Or, stating that the larger particles are more likely to be uptaken than the nanoparticles with less than 100 nm dia.? Can you quantify the difference in the rates of uptake? 

24) Finally, in both in-vitro comparison and biocompatibility tests, they should include the commercial Ga based contracting agents if possible.    

Author Response

Response to Reviewer 3 Comments

In Abstract:

Q1. In the other hand, the development of magnetic nanoparticles with a low dose, high image recognition and non-toxicity is currently a major challenge. "high image recognition" probably "high image contrast."

A1: Thanks for reviewer correction.  The word “high image recognition” is corrected to “high image contrast" on article.

Q2. The results show that the particle sizes of Ti- CoHA and TNT-CoHA are 418.6 nm and 127.5 nm, respectively. Although the particle size of TNT-CoHA is smaller than Ti-CoHA. However, the grain size of cobalt oxide in TNT-CoHA is larger than cobalt oxide in Ti-CoHA.  What does this mean? They should state which results give the larger (>100nm) particle size, and what results give the grain sizes to be different. In fact, particle size and grain size need to be defined.

A2: Thanks for reviewer suggestion.  We will modify the under description on article to make the expression more correct. "The particle sizes of Ti-CoHA and TNT-CoHA were 418.6 nm and 127.5 nm, respectively as observed using FE-SEM.  It is shown that CoHA can be obtained with a smaller particle size using a TNT electrode plate.  However, although the particle size of TNT-CoHA is smaller than Ti-CoHA.  The crystal size of the internal cobalt oxide of CoHA was calculated by XRD pattern.  The results indicate the crystal size of cobalt oxide in TNT-CoHA is larger than cobalt oxide in Ti-CoHA.  The TNT-CoHA has a larger crystal size of cobalt oxide makes the saturation magnetization (Ms) of TNT-CoHA to be 12.6 times higher than Ti-CoHA."

 Introduction

Q3. I suggest to use the term “contrast agent” instead of “developer” throughout the document.   

A3: Thanks for reviewer suggestion. The word "developer" is corrected to "contrast agent" on article.

 Q4. Introduction should perhaps start with the mechanism on how the nanoparticles enhance the MRI contrasts. 

A4: Thanks for reviewer suggestion.  We have made new fixes in Introduction.  The under sentences are adding to article.  "Since the magnetic dipole moment is induced in the magnetic nanoparticles under an applied magnetic field.  When the water molecules diffuse to the rim of the induced dipole moment, the magnetic relaxation process of the water protonhas interfered and the spin-spin relaxation time (T2) is shortened so that semaphore intensity of the water molecules in T2-weighted (T2WI) imaging is reduced [1].  This phenomenon causes magnetic nanoparticles as a negatively contrasted in T2WI magnetic resonance imaging, reducing the strength of tissue semaphore in the image andincrease the contrast of surrounding images [2,3].  Therefore the contrast agentcan increase contrast by magnetic nanoparticles [4]."

Q5."Frank SJ et al. pointed out that cobalt chloride has a paramagnetic property and is a good candidate for developers." This sentence should be followed with a reference to this work. "

A5: Thanks for reviewer correction.  The word "developer" is corrected to "contrast agent" on article.

Q6."Combined with Acetylcysteine pharmacy can be applied to MRI marker for prostate cancer treatment [14]." There is no subject in this sentence, or is "pharmacy" the subject indeed?

A6: Thanks for reviewer correction.  The word "pharmacy" is corrected to "acetylcysteine drug” on article.

Q7. Is hydroxyapatite anniversary applicable to all organs or mostly for bone tissues? 

A7: Thanks to reviewer for asking questions.  Hydroxyapatite (HA) is the main inorganic component of bones and teeth of vertebrates[5]. Mainly used in bone tissue engineering [6,7]. Recently, the synthesis of HA is easy to control and has excellent biocompatibility.  Therefore, there are also studies on the use of HA as a drug carrier in different organs, such as the liver [8].

2. MATERIALS AND METHODS  

Q8. "fig.1" should be "Fig.1"

A8: Thanks for reviewer suggestion.  Which has been revised in this article.

Q9. The crystallite sizes of CoHA, calculated by the Scherrer equation [22]:

So far, the authors used “particle size,” “grain size,” and “crystal size,” in the manuscript. They need to strictly define what these terms are throughout the manuscript.  

A9: Thanks for reviewer suggestion.  We have redefined these terms are throughout the article.  The under description was added in section 2.3.  “The morphology and particle size of TNT, Ti-CoHA and TNT-CoHA were observed by a field emission scanning electron microscope (FE-SEM) (JSM-7610F, JEOL, Tokyo, Japan).” “The crystal sizes of CoHA, calculated by the Scherrer equation [21]”

Q10. Strictly speaking, they should keep the same style in providing the manufacture/vendor information. Some are defined with country only, and the others are defined with City/Country.

A10: Thanks for reviewer suggestion.  We have made corrections in this article in the following format (model, company, city, country).

3. RESULTS AND DISCUSSION  

Q11. fig. 2 should be Fig. 2.

A11: Thanks for reviewer suggestion.  Which has been revised in article.

Q12. Along with the EDS spectra in Fig. 2, compositions obtained by EDS should be presented. The readers want to know the compositions of the impurities, such as F and C, as well as the Ti to O ratios for both samples. 

A12: We have added other compositions in Figure 2.

Fig.2 The surface morphology and composition of the (A, C) pure titanium and (B, D) TNT was observed by FE-SEM and EDS, respectively. The red frame is a magnified image of TNT.

Q13. "The Ti-CoHA and TNT-CoHA particles were 418.6 ±8.0 nm and 127.5 ±2.9 nm, respectively." Should mention the details of how these particle sizes were obtained. I think, you can see the smaller "grains" consisting of these "particles" by looking at the images at higher resolutions. 

A13: Thanks to Reviewer for asking questions.  We used Image-Pro Plus software to analyze FE-SEM images to obtain particle size. At the same time, the under description was added in the Materials and Methods section.  "In addition, the FE-SEM image was analyzed using Image-Pro Plus software (Media Cybernetics, Version 4, MD, USA) to obtain the particle size of the powder."

Q14. Need to discus why the two sets of data (EDS vs. IPC) on Co concentration are different here more quantitatively. What was the electron energy used in obtaining the EDS spectra which affects the depth of information (how deep were the X-rays collected by EDS to yield the Co concentration) under this condition. This will in tern, define what the authors claim to be "surface."

A14: Thanks to Reviewer for asking questions.  The ICP-OES analysis was performed after the powder was completely dissolved in 65% nitric acid.  So, we define ICP-OES as the overall composition.  In the other hand, EDS is an additional feature of FE-SEM and is often used to detect the composition of the surface of a material. X-rays are emitted throughout the volume of material into which the electron beam is scattered and decelerated, so the resolution is commonly 1 micrometer laterally and about 1 – 2 micrometers in depth (http://www.andersonmaterials.com/edx-eds.html).  The size of CoHA is nanometer grade, so it can only detect the elemental composition of the surface. In addition, we also provide raw data for TNT-CoHA samples for your reference (Fig S1.).

 Fig S1. The surface composition of the TNT-CoHA was obtaind by EDS.

 Q15. On Fig. 4, the entire XRD spectra (10-70degrees) should be given in addition to the narrow range scans given in Fig. 4A. At least in the ranges to show all the peaks mentioned in the body.    

A15: Thanks for reviewer suggestion.  which has been revised in Fig.4.

Fig.4 The crystal structure and chemical composition of Ti-CoHA and TNT-CoHA: (A) XRD patterns, (B) is a magnified image of XRD patterns and (C) ATR-FTIR spectra.

Q16. On Fig. 4 B, blue line spectrum (Ti-CoHA) is lower than Red line spectrum. Is this due to the difference in the amount of the powders photons traverse? If so, can you normalize them by using for example, 2 PO4lines, which would make the argument about the difference in the OH groups meaningless.

A16: Thank you for your question.  The peak value of FTIR will vary due to the content of specific functional groups. We tried to compare the two lines on the same baseline (green arrows and lines).  All functional groups of TNT-CoHA were higher than Ti-CoHA, indicating that the content of hydroxyl and phosphate was more.  In addition, the band of FTIR spectra represents the bond strength between functional groups. In this study, FTIR spectra were obtained by a Thermo Nicolet 380 spectrometer in the spectral range of 4000–650 cm−1and a resolution of 2 cm−1. The difference between TNT-CoHA and Ti-CoHA in the OH group is 27 cm−1(>2 cm−1).  Therefore, it is meaningful to discuss the differences in OH groups.

Fig.4 The crystal structure and chemical composition of Ti-CoHAand TNT-CoHA:(A) XRD patterns, (B) is a magnified image of XRD patterns and (C) ATR-FTIR spectra.

Q17. The cobalt oxide crystallite sizes of CoHA, calculated by the Scherrer equation [22], were about 2.7 nm for Ti-CoHA and 3.3 nm for TNT-CoHA, respectively.  No data (spectra shown for these XRD peaks. With the peaks shown in Fig. 4 A for another peak, the uncertainties in the FWHM may not be small. How confident are the authors on this data (3.3nm vs. 2.7nm)? Please give the estimated experimental error values.

A17: Thank you for your question.  Our numerical value is to directly read the data points on the XRD pattern.  However, the angle of the FWHM of the XRD pattern is not an integer.  Therefore, the error of 2θ angle is ± 0.02 degrees, and the error of θ angle is ± 0.01 degrees.  At the same time, we attach the original data for your reference (Table S1).  The crystal sizes of CoHA, calculated by the Scherrer equation [21]:

The FWHM is the full width at half maximum of the diffraction peak, θ is the angle of the diffraction peak.

Table S1. The construct parameters of the as synthesized CoHA particles by XRD analysis.

Sample

plane

2?

cos?

FWHM

Co crystal size (Xs) (nm)

Ti-CoHA

111

18.80

0.986572

0.52

2.705

TNT-CoHA

111

18.72

0.986686

0.43

3.271

 Q18. Fig. 5B; the results show that the crystal size of all magnetic nanoparticles is proportional to the magnetic field strength. I do not get this statement at all from Fig. 5 B.

A18: Thanks for reviewer question.  The x-axis coordinate of Fig. 5 B should be crystal size.  We have made changes.  From Fig. 5 B, it can be found that the magnetic field strength of the same magnetic nanoparticles increases with the increase of crystal size (dashed line).

Fig.5 (A) Mass magnetization of CoHA and TNT-CoHA by superconducting quantum interference device (SQUID). (B) Magnetic nanoparticle size effects on magnetic behavior (by XRD) ;( modify from Ref. MWCNT/CoFe2O [9], CoMoFe [10]and MgFe2O4[11], respectively.)

Q19. In summary, TNT-CoHA has a smaller particle size and a larger cobalt oxide size relative to Ti-CoHA This statement does not make sense either without the strict definitions of the particle/grain/crystal sizes. 

A19: Thanks for reviewer suggestion.  We have reconfirmed the definition of particle/grain/crystal sizes.

 3.4 In vitro MRI examination  

Q20. The relaxivity coefficient (r2) obtained from the gradient of the R2curve and the magnetic nanoparticle molar concentration is a standardized contrast intensity index. [42]. The higher the r2value, the lower the concentration of nanoparticle required in the same T2image. Are lower case r2s simple typing mistakes of R2, or does lower case r2mean something else?

A20: First, the sample is analyzed by MRI according to the parameter setting(matrix size = 256 x 256, field of view = 180 mm x 180 mm, slice thickness = 5 mm, echo time = 26 ms, repetition time = 100 ms, number of acquisitions = 2), and the computer process will automatically calculate T2.  Convert T2to R(1 / T2) value (For Fig. 6B 1/T2vs. concentration).  Finally, the relaxivity coefficient (r2) obtained from the gradient of the R2curve and the molar concentration of the magnetic nanoparticles [1].  The r2is calculated by the following formula(eq 1):[12]

   (eq 1)

where  is the observed relaxation time in the presence of CoHA,  is the relaxation rate of pure gelatin and  is the Co ion concentration.

T2: The value calculated by the computer in the MRI image.

R2: Converting the R2curve via (1 / T2) conversion.

r2: Relaxivity coefficient (r2) is an indicator of contrast efficiency.

 Q21. Calculating the overall surface area of the particles indicated that Ti-CoHA was 10.8 times higher than TNT-CoHA. That is to say, when the concentration of the particles is the same, the surface effect of TNT-CoHA is higher than that of Ti-CoHA.

Explain how this conclusion was obtained? Can you back it up by the experiments utilizing Surface Area Analyzer (such as Micromeritics ASAP 2020)? Are the author simply taking the average size of the particles of the two samples and comparing the surface areas of the same volume? Is assuming the concentrations of the two agents to be equal a reasonable assumption? Why not use the same weights for the two? Most importantly, can you quantitatively explain the difference shown in Fig. 6 with the increase in the surface area alone (I do recognize that the authors mention that many factors affect inn the results.)?

A21: Thanks to Reviewer for asking questions.  Sorry for our negligence, we changed the concentration to weight. "That is to say, when the weightsof the particles is the same, the surface effect of TNT-CoHA is higher than that of Ti-CoHA”.  In addition, we were unable to obtain a surface area analyzer for experiments. Therefore, we use the results of FE-SEM to calculate the surface area (Fig S2.). However, there are many factors affecting the relaxation efficiency coefficient such as particle size, crystal size, surface effect and saturation magnetic field strength [13].  In terms of surface effect, as suggested by Reviewer TNT-CoHA has a better negative contrast when the "weight" is the same (Fig 6A).  We also added a sentence to the article. "Therefore, in Fig 6A TNT-CoHA has a good negative contrast when the powder weight is the same".

Fig S2. Calculation of sample surface area

Fig.6 (A) T2-weighted MRI images of HA (commerce), CoHA and TNT-CoHA suspended in gelatin at different concentrations. (B) The T2relaxation rate R2(1/T2) against cobalt concentration of Co-HA. (C) Magnetic nanoparticle magnetization effects on relaxivity coefficient r2;( modify from Ref. MWCNT/CoFe2O[9], CoFe [14], Fe3O4[15]and FeHA[8], respectively.

Q22. The literature indicates that nanoparticles with a diameter greater than 100 nm are prone to cellular uptake [47]. Are you stating that nanoparticles less than 100 nm in diameter are not likely to be uptaken by cells? Or, stating that they are larger particles are more likely to be uptaken that the nanoparticles with less than 100 nm dia.? Can you quantify the difference in the rates of uptake? 

A22: Thanks to Reviewer, first of all our references have errors in importing. We have now modified it [1].  In addition, we did not say that nanoparticles smaller than 100 nm are less likely to be absorbed by cells.  We mean that nanoparticles smaller than 100 nanometers are easier to shuttle between cells than nanoparticles larger than 100 nanometers. Therefore, nanoparticles larger than 100 nm are easily retained inside the cell to be absorbed.

 Q23. The literature indicates that nanoparticles with a diameter greater than 100 nm are prone to cellular uptake [47]. Are you stating that nanoparticles less than 100 nm in diameter are not likely to be uptaken by cells? Or, stating that the larger particles are more likely to be uptaken than the nanoparticles with less than 100 nm dia.? Can you quantify the difference in the rates of uptake? 

A23: As with the Q22 topic, we have already responded.

 Q24. Finally, in both in-vitro comparison and biocompatibility tests, they should include the commercial Ga based contracting agents if possible.

A24: Many thanks to Reviewer for their advice. We know that the material part of the current contrast agent contains Ga.  However, the use of Ga has potential doubts about the human body, which is a known fact [16]. Therefore, we did not use Ga as a control group for biocompatibility testing.

 References

1.         Jun, Y.W.; Lee, J.H.; Cheon, J. Chemical design of nanoparticle probes for high-performance magnetic resonance imaging. Angew Chem Int Ed Engl 2008,47, 5122-5135.

2.         Kalambur, V.S.; Han, B.; Hammer, B.E.; Shield, T.W.; Bischof, J.C. In vitro characterization of movement, heating and visualization of magnetic nanoparticles for biomedical applications. Nanotechnology 200516, 1221.

3.         Kalambur, V.S.; Han, B.; Hammer, B.E.; Shield, T.W.; Bischof, J.C. In vitrocharacterization of movement, heating and visualization of magnetic nanoparticles for biomedical applications. Nanotechnology 200516, 1221-1233.

4.         Key, J.; Cooper, C.; Kim, A.Y.; Dhawan, D.; Knapp, D.W.; Kim, K.; Park, J.H.; Choi, K.; Kwon, I.C.; Park, K., et al.In vivo nirf and mr dual-modality imaging using glycol chitosan nanoparticles. J Control Release 2012163, 249-255.

5.         Bohner, M.; Galea, L.; Doebelin, N. Calcium phosphate bone graft substitutes: Failures and hopes. Journal of the European Ceramic Society 201232, 2663-2671.

6.         Wang, Y.; Yang, X.; Gu, Z.; Qin, H.; Li, L.; Liu, J.; Yu, X. In vitro study on the degradation of lithium-doped hydroxyapatite for bone tissue engineering scaffold. Mater Sci Eng C Mater Biol Appl 201666, 185-192.

7.         Park, S.A.; Lee, S.H.; Kim, W.D. Fabrication of porous polycaprolactone/hydroxyapatite (pcl/ha) blend scaffolds using a 3d plotting system for bone tissue engineering. Bioprocess Biosyst Eng 2011,34, 505-513.

8.         Adamiano, A.; Iafisco, M.; Sandri, M.; Basini, M.; Arosio, P.; Canu, T.; Sitia, G.; Esposito, A.; Iannotti, V.; Ausanio, G., et al.On the use of superparamagnetic hydroxyapatite nanoparticles as an agent for magnetic and nuclear in vivo imaging. Acta Biomater 201873, 458-469.

9.         Wu, H.; Liu, G.; Wang, X.; Zhang, J.; Chen, Y.; Shi, J.; Yang, H.; Hu, H.; Yang, S. Solvothermal synthesis of cobalt ferrite nanoparticles loaded on multiwalled carbon nanotubes for magnetic resonance imaging and drug delivery. Acta Biomater 20117, 3496-3504.

10.       MOHAMMAD, A.; RIDHA, S.; MUBARAK, T. Structural and magnetic properties of mg-co ferrite nanoparticles. Digest Journal of Nanomaterials & Biostructures (DJNB) 201813.

11.       Druc, A.; Dumitrescu, A.; Borhan, A.; Nica, V.; Iordan, A.; Palamaru, M. Optimization of synthesis conditions and the study of magnetic and dielectric properties for mgfe2o4 ferrite. Open Chemistry 201311.

12.       Joshi, H.M.; Lin, Y.P.; Aslam, M.; Prasad, P.V.; Schultz-Sikma, E.A.; Edelman, R.; Meade, T.; Dravid, V.P. Effects of shape and size of cobalt ferrite nanostructures on their mri contrast and thermal activation. J. Phys. Chem. C 2009, 17761–17767.

13.       Shokrollahi, H. Contrast agents for mri. Materials Science and Engineering: C 201333, 4485-4497.

14.       Joshi, H.M.; Lin, Y.P.; Aslam, M.; Prasad, P.; Schultz-Sikma, E.A.; Edelman, R.; Meade, T.; Dravid, V.P. Effects of shape and size of cobalt ferrite nanostructures on their mri contrast and thermal activation. The Journal of Physical Chemistry C 2009113, 17761-17767.

15.       Jun, Y.-w.; Huh, Y.-M.; Choi, J.-s.; Lee, J.-H.; Song, H.-T.; Kim, S.; Kim, S.; Yoon, S.; Kim, K.-S.; Shin, J.-S. Nanoscale size effect of magnetic nanocrystals and their utilization for cancer diagnosis via magnetic resonance imaging. Journal of the American Chemical Society 2005,127, 5732-5733.

16.       Neuwelt, E.A.; Hamilton, B.E.; Varallyay, C.G.; Rooney, W.R.; Edelman, R.D.; Jacobs, P.M.; Watnick, S.G. Ultrasmall superparamagnetic iron oxides (uspios): A future alternative magnetic resonance (mr) contrast agent for patients at risk for nephrogenic systemic fibrosis (nsf)? Kidney Int 200975, 465-474.

Round  2

Reviewer 1 Report

The quality of the paper is improve after answering to reviewer's questions. The English correction were done and now the paper is more readable.

Author Response

Response to Reviewer 1 Comments

Q1: The quality of the paper is improve after answering to reviewer's questions. The English correction were done and now the paper is more readable.

A1: Thank you for your comments. let article more complete and more readable.

Reviewer 2 Report

There is some improvement in the revised text, however two of my points from previous review were not really addressed. So, please address the following issues in the text of the submission:

1. The explanation of the way to obtain T2 values is not satisfactory. Using spin-echo pulse sequence you need at least two (and preferably more) images obtained with different echo times (or time for multiple echo train) in order to calculate single T2.  In the description in subsection 2.4 you indicate single echo time only (i.e. 26 ms). So please add information in the text how many images corresponding to different echo times did you use (and what were values of these echo times ?) to calculate T2 values.  Moreover, I suggest to replace “spin echo imaging sequencing” with “spin echo imaging pulse sequence”

2. Description of Fig. 5B in subsection 3.4 is still not correct. If the data in Fig. 5B are correct then it is not true that T2 for TNT-CoHA is higher (or “superior”) than T2 for Ti-Co-HA. It is opposite. Look at the graph which you present in Fig. 5.B, where you show dependence of 1/T2 (not T2 !) and make relevant correction to the statement in the text.

Moreover, some language improvement is still recommended (example e.g.  “MRI developer” term still left in Conclusion).

Author Response

Response to Reviewer 2 Comments

Q1: The explanation of the way to obtain T2 values is not satisfactory. Using spin-echo pulse sequence you need at least two (and preferably more) images obtained with different echo times (or time for multiple echo train) in order to calculate single T2.  In the description in subsection 2.4 you indicate single echo time only (i.e. 26 ms). So please add information in the text how many images corresponding to different echo times did you use (and what were values of these echo times ?) to calculate T2 values.  Moreover, I suggest to replace “spin echo imaging sequencing” with “spin echo imaging pulse sequence”

A1: Thank you for your suggestion. We have added under sentences to materials and methodssection.  “We used a customized multiple-spin-echo pulse sequence to obtain T2 values.  It was developed to acquire 8 spin echoes with an optional choice of echo spacing (ESP) from a minimum of 8.9ms.  Parameters of the multi-echo imaging protocol are listed as follows: TR=4000ms, TE=8.9, 17.8, 26.6, 35.5, 44.4, 53.3, 62.2, 71.0ms, slice thickness= 5mm and number of acquisition= 2.”

 Q2: Description of Fig. 5B in subsection 3.4 is still not correct. If the data in Fig. 5B are correct then it is not true that T2for TNT-CoHA is higher (or “superior”) than T2for Ti-Co-HA. It is opposite. Look at the graph which you present in Fig. 5.B, where you show dependence of 1/T2(not T2!) and make relevant correction to the statement in the text.

A2: Thank you, we have made corrections. The ordinate in Fig.6B is “R2”, but in section 3.4, we use “T2” as a narrative.  However, as you said, we found this to be wrong.  Because the value of R2is (1 / T2). Therefore, we modify it in this article and Fig.6B.  “The spin-spin relaxivity R2(R2=1/T2) is usually used to indicate of paramagnetic effect.  The higher the R2, the shorter the T2relaxation time, the darker the tissue image, and the larger the contrast effect compared with the tissue with lower R2.  In addition, the R2value of TNT-CoHA (46) was superior tothat of Ti-CoHA (33) at a cobalt ion concentration of 0.12 Mm (Fig.6B).  Previous studies have shown that Ms is proportional to the value of R2[1-3]. Therefore, TNT-CoHA has a good T2image effect.”

 Fig.6 (B) The T2 relaxation rate R2 (1/T2) against cobalt concentration of Co-HA.

 Q3: Moreover, some language improvement is still recommended (example e.g.  “MRI developer” term still left in Conclusion).

A3: Thank you for review, we have made corrections.

 References

1.         Shokrollahi, H. Contrast agents for mri. Mater Sci Eng C Mater Biol Appl 201333, 4485-4497.

2.         Shin, T.H.; Choi, Y.; Kim, S.; Cheon, J. Recent advances in magnetic nanoparticle-based multi-modal imaging. Chem Soc Rev 201544, 4501-4516.

3.         Kim, J.S.; Kuk, E.; Yu, K.N.; Kim, J.-H.; Park, S.J.; Lee, H.J.; Kim, S.H.; Park, Y.K.; Park, Y.H.; Hwang, C.-Y. Antimicrobial effects of silver nanoparticles. Nanomedicine: Nanotechnology, Biology and Medicine 20073, 95-101.

Reviewer 3 Report

I appreciate the responses from the author, and the improvements have been made. However, I have a few more inquiries.
1) There should be a sentence or two to clarify when and how the powders are removed from the substrate electrodes. Do they have the same particle sizes as produced on the substrates and removed from the substrates by FESEM observations?
2) On Fig. 4, you do not need B. Just take XRD spectrum B out and plot A (entire scan) twice as wide. (I suppose the old XRD and FTIR plots on page 28 were left by mistake?)
3) On page 16: Therefore, we calculated the crystal size by the position of cobalt oxide (18.6°) in the XRD pattern. The cobalt oxide crystallite sizes of CoHA, calculated by the Scherrer equation[26]The cobalt oxide crystal sizes of CoHA, calculated by the Scherrer equation [26], were about 2.7 nm for Ti-CoHA and 3.3 nm for TNT-CoHA, respectively.
I hardly see any peaks at 18.6 degrees on the XRD spectra of the two samples given in Fig. 5. Can you show actual fits and the spectra at this region around 18.6 deg.?
With such small peaks, the uncertainties in determining the FWHM values should be large. Therefore the difference between 2.7 nm and 3.3 nm is most likely negligible. Unless the authors can show the errors (or standard deviations) of the FWHM are negligible compared such small crystal sizes obtained, I cannot agree with the conclusion that they have different crystal sizes.
4) On Page 18: Calculating the overall surface area of the particles indicated that Ti-CoHA was 10.8 times higher than TNT-CoHA. That is to say, when the concentrationweights of the particles is the same, the surface effect of TNT-CoHA is higher than that of Ti-CoHA.
  These two sentences seem to contradict each other. Smaller that the particle diameter, the proportional ratio of the atoms at surfaces increases. Thus, the smaller particles have larger surface area, given the same weight (the same number of atoms). Need to be re-written.   

Author Response

Response to Reviewer 3 Comments

Q1: There should be a sentence or two to clarify when and how the powders are removed from the substrate electrodes. Do they have the same particle sizes as produced on the substrates and removed from the substrates by FESEM observations? 

A1: Thank you for your suggestion.  In Materials and Methodssection, we have added procedure to remove the powder from the electrode plate as described below.  “After the reaction, use a plastic spoon to remove the powder from the electrode plate”.  In addition, effect of collection process on the morphology of the powder was observation by FESEM.  No effect on the powder morphology on before and after the collection (Fig. S1).

Fig.S1 Effect of the removal procedure on Ti-CoHA powder morphology was observed by FESEM.  (a) powder morphology on electrode and (b) collected powder morphology through removal procedure.

Q2: On Fig. 4, you do not need B. Just take XRD spectrum B out and plot A (entire scan) twice as wide. (I suppose the old XRD and FTIR plots on page 28 were left by mistake?)

A2: Thank you for your suggestion.  Figure is corrected according to your suggestion.

Fig.4 The crystal structure and chemical composition of Ti-CoHA and TNT-CoHA: (A) XRD patterns, (B) FTIR spectra.

Q3: On page 16: Therefore, we calculated the crystal size by the position of cobalt oxide (18.6°) in the XRD pattern. The cobalt oxide crystallite sizes of CoHA, calculated by the Scherrer equation[26]The cobalt oxide crystal sizes of CoHA, calculated by the Scherrer equation [26], were about 2.7 nm for Ti-CoHA and 3.3 nm for TNT-CoHA, respectively. I hardly see any peaks at 18.6 degrees on the XRD spectra of the two samples given in Fig. 5. Can you show actual fits and the spectra at this region around 18.6 deg.?

A2: Thank you for your suggestion.  We have added a magnified XRD spectrum for your consultation (Fig.S2).

Fig.S2 The crystal structure of Ti-CoHA and TNT-CoHA XRD patterns (16-20°).

Q4:With such small peaks, the uncertainties in determining the FWHM values should be large. Therefore the difference between 2.7 nm and 3.3 nm is most likely negligible. Unless the authors can show the errors (or standard deviations) of the FWHM are negligible compared such small crystal sizes obtained, I cannot agree with the conclusion that they have different crystal sizes. 

A4:Thank you for your suggestion.  We bring the deviation of the values into the size of the crystal.  Using statistical software for analysis, we can get results that are significantly different (Fig S3).

 Fig.S3 The size of the crystal of cobalt-substituted hydroxyapatite by XRD data calculation

Q5: On Page 18: Calculating the overall surface area of the particles indicated that Ti-CoHA was 10.8 times higher than TNT-CoHA. That is to say, when the concentrationweights of the particles is the same, the surface effect of TNT-CoHA is higher than that of Ti-CoHA. These two sentences seem to contradict each other. Smaller that the particle diameter, the proportional ratio of the atoms at surfaces increases. Thus, the smaller particles have larger surface area, given the same weight (the same number of atoms). Need to be re-written.  

A5:Thank you for your suggestion.  We changed the original sentence to “Calculating the surface area of single particle indicated that Ti-CoHA was 10.8 times higher than TNT-CoHA. Therefore, when the weights of the particles are the same, the surface effect of TNT-CoHA is higher than that of Ti-CoHA.”
